# Collaborative Optimization Method of Power and Efficiency for LCC-S Wireless Power Transmission System

Ming Xue [1,2,*], Qingxin Yang [1], Chunzhi Li [2], Pengcheng Zhang [1], Shuting Ma [2] and Xin Zhang [2]

[1] State Key Laboratory of Reliability and Intelligence of Electrical Equipment, Hebei University of Technology, Tianjin 300132, China; qxyang@tjut.edu.cn (Q.Y.); 201611401014@stu.hebut.edu.cn (P.Z.)
[2] Tianjin Key Laboratory of Advanced Electrical Engineering and Energy Technology, Tiangong University, Tianjin 300387, China; lcz15030712832@163.com (C.L.); mashuting16@163.com (S.M.); zhangxin@tiangong.edu.cn (X.Z.)
\* Correspondence: xueming@tiangong.edu.cn; Tel.:+86-159-0222-4514

**Abstract:** Dynamic wireless charging enables moving equipment such as electric vehicles, robots to be charged in motion, and thus is a research hotspot. The applications in practice, however, suffer from mutual inductance fluctuation due to unavoidable environmental disturbances. In addition, the load also changes during operation, which makes the problem more complicated. This paper analyzes the impacts of equivalent load and mutual inductances variation over the system by LCC-S topology modeling utilizing two-port theory. The optimal load expression is derived. Moreover, a double-sided control strategy enabling optimal efficiency and power adjustment is proposed. Voltage conducting angles on the inverter and rectifier are introduced. The simulation and experimental results verify the proposed method.

**Keywords:** dynamic wireless power transmission; parameter identification; particle swarm algorithm; efficiency optimization; power control



## 1. Introduction

Nowadays, with the increasing degree of electrification in people's life, electric energy has become indispensable. The transmission medium used in traditional power transmission generally adopts metal wires and cables, which will have problems such as transmission loss, line aging, tip discharge, and so on. Therefore, human beings continue to explore new ways of electric energy transmission, and radio energy transmission technology came into being. Wireless power transfer technology was first proposed by Nikola Tesla, a famous electrical engineer in the middle and late 19th century [1]. It is a transmission mode to transfer electric energy from the power terminal to electrical equipment with the help of space invisible soft media (such as electric field, magnetic field, acoustic wave, etc.). Compared with the traditional way of using cable to transmit electric energy, this transmission mode is more safe, convenient, and reliable. It is considered to be a revolutionary progress in energy transmission and access.

Since the academic achievements of the Massachusetts Institute of Technology in 2007, which used the principle of magnetic coupling resonance to isolate and light 60 W bulbs [2], a large number of scholars have made in-depth theoretical basic research on the technology of magnetic coupling resonance radio energy transmission [3,4], and based on the prominent problems in different application scenarios such as electronic equipment [5], transportation [6], medicine [7], medical [8], and aviation [9]. Feasible proposals are presented from different dimensions. At present, the wireless power transmission (WPT) has been industrialized and popularized in the field of low-power electronic equipment, and several demonstration projects have been established in the field of transportation such as high-power electric vehicles [10] and railroad trains [11], which proved the feasibility of the technology. Although great progress has been made in the development of radio

power transmission technology, its maturity in stability, reliability, and efficiency, which cannot support its application in various fields, and further research is needed.

In a dynamic wireless power supply system, the coupling mechanism has a large space magnetic flux leakage [12], resulting in a low transmission efficiency of the system, and the received power fluctuation caused by environmental disturbance, coil offset, and the switching process of adjacent coil excitation and conduction is an important problem that restricts the industrial application. Therefore, it is necessary to consider the whole process of system operation as well as the actual working conditions. Then, the output characteristics of the magnetically coupled topology are analyzed by establishing a mathematical model of the system, and a cooperative control strategy that takes into account the transmission efficiency and the received power of the system is investigated so as to improve the transmission performance of the system and ensure the safe and reliable operation of the system.

In reference [13], an efficiently optimal method to improve the system efficiency while regulating the output voltage is proposed. Two overlapped DD coils are employed to generate an enhanced magnetic field without blind spots. An optimal current control method is proposed to adjust the direction and the ratio of the two transmitter currents as the receiver moves along the track, dynamically. In recent years, the so-called DD coils have been proposed [14]. They are conveniently used for the pickups of both stretched and lumped track dynamic wireless charge systems and, in the latter case, they are also used in the track [15]. In reference [16], the constant current (CC) and constant voltage (CV) charging strategies based on phase shift control of a controllable rectifier circuit and efficiency optimization strategy based on dynamic equivalent impedance matching are proposed, which can improve the WPT system transmission efficiency while realizing CC and CV charging. References [17,18] improve the performance of the system by optimizing the structure of the system. In reference [17], a method of harmonic elimination by adding a second-order liquidity coverage ratio (LCR) filter to the resonant part is studied to improve the output power and efficiency of the system. In reference [18], the primary compensation capacitance is well designed to regulate the transfer power fluctuation in the WPT system. However, this method will lead to a more complex system structure. References [19,20] improve the efficiency of IPT system by reasonably selecting the system parameters. Reference [19] considers the compensation of the input impedance of the rectifier bridge when designing the secondary compensation capacitance. Reference [20] designs capacitance inductance parameters which are insensitive to load changes. References [21–28] track and optimize the control of the system efficiency in real time during system operation. References [21–24] use global search to optimize the system efficiency, which can ultimately optimize the efficiency of the system, but the search process is usually slow. In reference [25], from the angle of balancing the original secondary current, the system efficiency is optimized by controlling the ratio of the original secondary current to the semi-controlled rectifier bridge. Reference [26] investigates the method of impedance transformation of a full-control rectifier bridge cascade with an additional direct current/direct current (DC/DC) circuit to load and analyses the effect of impedance transformation on efficiency. Reference [27] presents a maximum efficiency tracking method with integrated dynamic coupling coefficient estimation. The DC/DC converter is connected to the transceiver and the wireless communication device which is used for data transformation. When the coupling factor and load change, the system can track the maximum efficiency point in real time and control the output on the basis of identifying the coupling coefficient. However, a full-control bridge and DC/DC circuit increase the power conversion circuit and switching devices, and increase the system loss and complexity. Reference [28] presents a two-sided phase shift control method with an auxiliary measuring coil, which can achieve maximum efficiency transmission by adjusting the equivalent resistance. However, this method also increases the loss and complexity of the system.

From the above references, research on efficiency optimization and the constant voltage and current control of a dynamic wireless power supply system has been carried

out from a single performance index, while considering the transmission efficiency and receiving power of wireless power supply system; there are few studies on the collaborative control between them. Therefore, based on the actual working conditions of the dynamic wireless power transmission system, and by analyzing the output characteristics of the compensation topology, a strategy to control the output power with maximum efficiency tracking is presented. Firstly, the LCC-S topological mathematical model is established using two-port theory, and the influence of equivalent load, mutual inductance, and transmission side compensation inductance on the transmission performance of the system is analyzed. Secondly, around the fluctuation of power and efficiency during the operation of the system, each voltage conduction angle is determined in the inverter circuit and the semi-controlled rectifier circuit. In the adjustable range of mutual inductance, the system efficiency optimization and stable output power output control are achieved by adjusting the degrees of freedom of control on the transmitter-side and receiver-side. Both simulation and experimentation can verify the feasibility of the proposed bilateral control strategy.

## 2. Analysis of Mathematical Model of LCC-S Compensation Topology

### 2.1. Theoretical Analysis

The dynamic wireless power supply system studied in this paper adopts a segmented rail structure. During the movement of the electric vehicle, the transmitting unit is switched on and off in sequence through the external sense and control circuit. Only the single transmitter unit located under the electric vehicle is turned on. When the transmitting unit obtains the turn-on signal, the power frequency alternative current passes through the rectifier, inverter and compensation circuit, and the high frequency alternative current is sent to the transmitting coil. Thus, a high-frequency magnetic field is generated in space. The electric energy obtained by the receiving coil that relies on the coupling of the magnetic field to supply power to the electric vehicle through the conversion circuit. Figure 1 shows the structure diagram of the segmented rail type dynamic wireless power supply system.

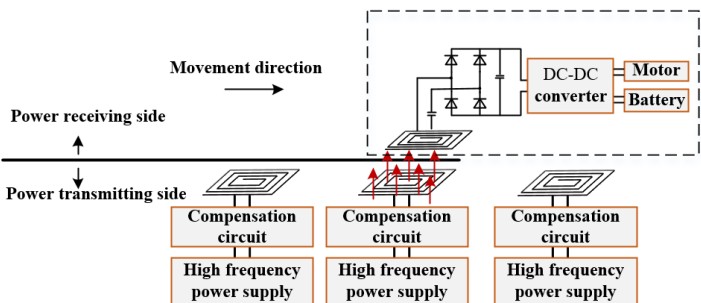

**Figure 1.** Schematic diagram of the segmented rail type dynamic wireless power supply system.

In the WPT, the coil is equivalent to a large inductive resistance. By adding capacitance to the primary side for capacitive compensation, the primary equivalent circuit load can be improved, thereby increasing the power factor at the input and reducing the apparent power. In the secondary equivalent circuit, the load current is strongly influenced by the coil inductance. By compensating the capacitance of the secondary circuit, the load output capacity can be increased to achieve the maximum power output. By adding a resonant compensation network to compensate the reactive power of the inductance of the loosely coupled transformer coil, the system is resistive to reduce the reactive power capacity while reducing the voltage and current stress of each power device in the power converter and improving the system transmission efficiency. In this paper, lcc-s compensation topology is adopted; that is, LCC compensation topology is adopted on the transmitting side, and S compensation topology is adopted on the receiving side. The equivalent circuit diagram is shown in Figure 2, where $U_1$ is the equivalent circuit power supply voltage, $I_f$ is the inverter output current, $I_1$ and $I_2$ are the currents of the resonant coil on the transmitting and receiving side, $L_f$ is the inductance of the transmitting side compensation coil, $L_1$ and $L_2$ are

the inductance of the transmitting and receiving side resonant coil, $C_f$ is transmitting side series compensation capacitor, $C_1$ is the transmitting side parallel compensation capacitor, $C_2$ is the receiving side series compensation capacitor, $M$ is the coil mutual inductance, $R_f$ is the internal resistance of the transmitting side compensation inductor, $R_1$ and $R_2$ are the internal resistance of the transmitting and receiving side resonant coil, $R_{Leq}$ is equivalent load resistance, the system working angular frequency is $\omega$. $L_f$, $C_f$ and $C_1$ form the LCC-type compensation topology, and $L_2$ and $C_2$ form the S-type compensation topology.

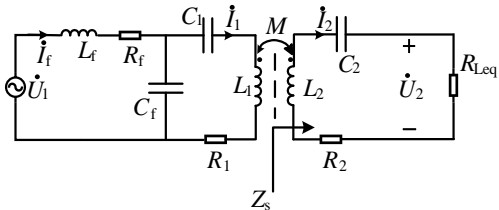

**Figure 2.** LCC-S type compensation topology equivalent circuit diagram.

In this paper, the LCC-S compensation topology is used, which can not only achieve constant voltage output at the resonance point but can also realize the cross-current characteristics of the transmitting coil, balance the stress of the component voltage and current, and has better misalignment tolerance. In addition, since the dynamic wireless power supply system study has only one transmitter at a time, the single-transmit–single-receive structure is used as the research object in the establishment of the mathematical model.

This paper adopts an array of charging coils on the transmitting side. Details on the proposed receiver coil design are provided later. The transmitter coils are fixed under the roadway while the receiver coils are free to move along the path of the transmission, and power is delivered through the gaps of the coils.

It can be seen from Figure 2 that the impedance of the receiving side circuit is expressed as:

$$Z_s = R_2 + R_{Leq} + j\omega L_2 + \frac{1}{j\omega C_2} \tag{1}$$

To facilitate the calculation, map the receiving side circuit to the transmitting side to obtain the equivalent circuit to the transmitting side, as shown in Figure 3.

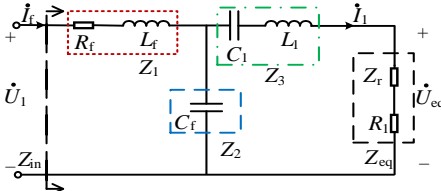

**Figure 3.** LCC-S type primary side equivalent circuit.

The reflection impedance equivalent to the transmitting side in Figure 3 is:

$$Z_r = \frac{\omega^2 M^2}{Z_s} \tag{2}$$

Use the two-port theory to establish the mathematical model of the primary side as follows:

$$\begin{bmatrix} \dot{U}_1 \\ \dot{I}_f \end{bmatrix} = \begin{bmatrix} A_{11} & A_{12} \\ A_{21} & A_{22} \end{bmatrix} \cdot \begin{bmatrix} \dot{U}_{eq} \\ \dot{I}_1 \end{bmatrix} \tag{3}$$

where $A_{11}$, $A_{12}$, $A_{21}$, and $A_{22}$ are the parameters of the two-port $A$.

For ease of description, this paper uses a T-shaped network instead of the compensation topology of the primary side, where:

$$A = \begin{bmatrix} A_{11} & A_{12} \\ A_{21} & A_{22} \end{bmatrix} = \begin{bmatrix} \frac{Z_1+Z_2}{Z_2} & \frac{P}{Z_2} \\ \frac{1}{Z_2} & \frac{Z_2+Z_3}{Z_2} \end{bmatrix} \tag{4}$$

$$P = Z_1Z_2 + Z_2Z_3 + Z_1Z_3 \tag{5}$$

$$Z_1 = R_f + j\omega L_f \tag{6}$$

$$Z_2 = \frac{1}{j\omega C_f} \tag{7}$$

$$Z_3 = j\omega L_1 + \frac{1}{j\omega C_1} \tag{8}$$

From Equation (2) to Equation (8), the system input impedance can be obtained:

$$Z_{in} = R_f + \frac{A_{11}Z_{eq} + A_{12}}{A_{21}Z_{eq} + A_{22}} \tag{9}$$

When the system is in the resonant state, the total impedances of the transmitter side and receiver side are presented as purely resistive, the power factor is maximized, and the resonant angular frequency of the transceiver circuit is consistent with the working angular frequency of the system $\omega$. At this time, the corresponding inductance and capacitance value should meet the following relationship:

$$\begin{cases} \omega L_f = \frac{1}{\omega C_f} \\ \omega L_2 = \frac{1}{\omega C_2} \\ \frac{1}{\omega C_1} = \omega(L_1 - L_f) \end{cases} \tag{10}$$

From Equation (10), we can obtain the current at the receiving and transmitting side under resonance state

$$\dot{I}_f = \frac{\dot{U}_1}{Z_{in}} \tag{11}$$

$$\dot{I}_1 = \frac{\dot{U}_1}{A_{11}Z_{eq} + A_{12}} = \frac{-U_1 L_f Z_s \omega j}{R_f M^2 \omega^2 + Z_s(R_1 R_f + L_f{}^2 \omega^2)} \tag{12}$$

$$\dot{I}_2 = \frac{\dot{I}_1 j\omega M}{Z_s} = \frac{U_1 L_f M \omega^2}{R_f M^2 \omega^2 + Z_s(R_1 R_f + L_f{}^2 \omega)} \tag{13}$$

The output power and transmission efficiency of the system are as follows:

$$P_o = \frac{\omega^4 L_f^2 M^2 R_{Leq} U_1{}^2}{\left[R_f M^2 \omega^2 + (R_2 + R_{Leq})(\omega^2 L_f^2 + R_1 R_f)\right]^2} \tag{14}$$

$$\eta = \frac{\omega^4 L_f^2 M^2 R_{Leq}}{R_s(\omega^4 L_f^2 + 2\omega^2 M^2 R_1 R_f) + R_f M^4 \omega^4 + R_s^2(\omega^2 L_f^2 R_1 + R_1^2 R_f)} \tag{15}$$

where $R_s = R_2 + R_{Leq}$. $P_o$ is the output power; and $\eta$ is the transmission efficiency of the system, which is the ratio of the input power of the resonant network on the transmitting side to the power at both ends of the load on the receiving side.

## 2.2. Analysis of Influencing Factors of System Power and Efficiency

In order to visually reflect the results of equivalent load and mutual inductance parameters on power and efficiency, Equations (14) and (15) are simulated using MATLAB software. Here, the effective value of AC power voltage is set to 50 V, the internal resistance

of the transceiver coil is 0.13 Ω, and the system resonance frequency f is 50 kHz, as shown in Figure 4.

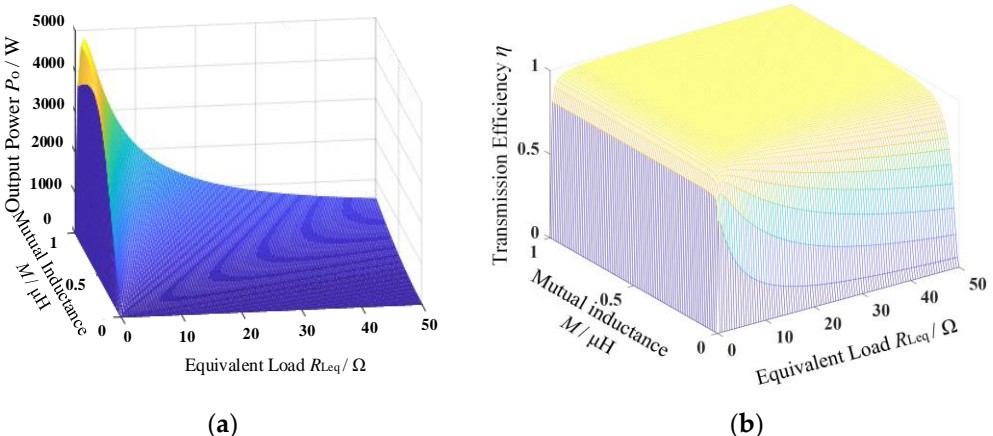

**Figure 4.** The relationship between $R_{\text{Leq}}$, $M$, and $P_{\text{o}}$, $\eta$. (**a**) The relationship between $R_{\text{Leq}}$, $M$, and $P_{\text{o}}$ and (**b**) the relationship between $R_{\text{Leq}}$, $M$, and $\eta$.

In Figure 4a, when the equivalent load value is small, the output power increases rapidly with the increase in mutual inductance. When the equivalent load continues to increase, the output power increases with the increase in mutual inductance, and the speed of increase decreases with the increase in load. In Figure 4b, when the mutual inductance value is small, the transmission efficiency of the system will first increase and then decrease with the increase in the equivalent load. Therefore, in order to take into account the transmission efficiency and output power of the system, the range of mutual inductance parameters can be estimated according to the size of the equivalent load.

### 3. Research on Control Strategy of Efficacy System

*3.1. Optimal Equivalent Load Analysis under Maximum Efficiency*

During the operation of a radio power transmission system, the equivalent load corresponding to the maximum transmission efficiency is the optimal equivalent load value. The value can be derived from the Equation (15) and given a derivative of 0, as follows:

$$\frac{d\eta}{dR_{\text{Leq}}} = 0 \tag{16}$$

The optimal equivalent load resistance can be determined by Equation (16):

$$R_{\text{best}} = \frac{\sqrt{R_1(L_f{}^2\omega^2 + R_1 R_f)(M^2\omega^2 + R_1 R_2)(R_2 L_f{}^2\omega^2 + R_f M^2\omega^2 + R_f R_1 R_2)}}{R_1 L_f{}^2\omega^2 + R_f R_1{}^2} \tag{17}$$

Equation (17) shows that when the system is disturbed by external factors, mutual inductance and load change, the equivalent load will usually change, and the system load will not be in the optimal value, resulting in inefficient operation. In addition, the parameters of most circuit elements in the system are fixed during the system operation, optimizing the circuit parameters that cannot maintain the high efficiency of the system under different loads. Therefore, in order to achieve the maximum efficiency tracking of the system, the equivalent resistance of the load must be controlled so that it is always equal to the optimal equivalent resistance value.

*3.2. Transmission Efficiency and Receiving Power Control Strategy*

In the actual wireless power supply system, an uncontrolled rectifier circuit is usually added to the receiving side, which is used to convert high-frequency AC to DC for the load power supply. However, the maximum efficiency tracking strategy proposed in this paper

needs to make the AC equivalent resistance of the rectifier bridge equal to the optimal load resistance value. The best way is to add a degree of control freedom on the receiving side. Therefore, the half-controlled rectifier circuit is used on the receiving side, and two MOSFET switching devices are used instead of the two diodes of the lower arm of the traditional uncontrolled rectifier circuit. The conduction angle of the rectifier bridge is introduced to meet the requirements of the degree of freedom and avoid adding extra power circuit. In addition, in order to take into account the stability of the output power, the conduction angle of the inverted circuit is added as another degree of control freedom. By introducing the phase-shifting angle of the inverted full bridge at the transmitter side, the variation law of the output voltage of the full bridge circuit is changed, and the control flexibility is improved to achieve the control of the output power. Additionally, the phase shift control method is used on the emitter side can reduce the influence of current stress on the system to a certain extent and reduce the device loss.

Figure 5 shows the WPT system of a semi-controlled rectifier bridge, where $U_{in}$ is the DC voltage, $u_1$ and $u_2$ are the inverting bridge output voltage and rectifier bridge input voltage, $i_f$ is the emitter side output current, $i_2$ is the receiving side input current, $R_L$ is the AC load resistance, and $U_o$ and $I_o$ are the DC output voltage and current, respectively.

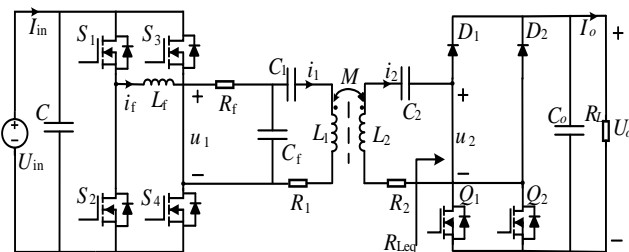

**Figure 5.** WPT system based on half-controlled rectifier bridge.

Assuming that all the switches and diodes are ideal components and the system operates in a resonant state, the control signal of the system circuit and the corresponding current and voltage waveform can be obtained as shown in Figure 6.

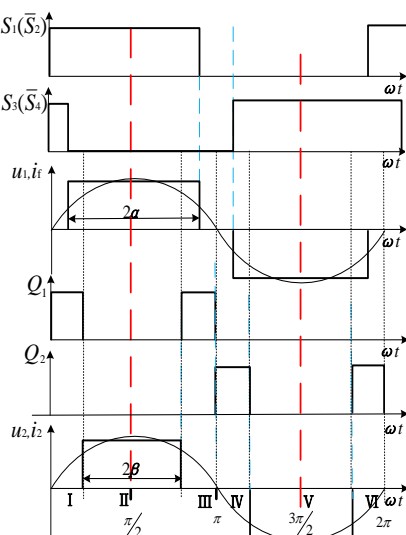

**Figure 6.** System control signal waveform values.

In Figure 6, $2\alpha$ and $2\beta$ are the voltage conduction angles of the inverting circuit and the semi-controlled rectifier circuit, respectively. According to the structure of the LCC-S resonant compensation circuit, $u_1$ and $u_2$ have the same phase. In the inverter circuit, $S_3$ is ahead of $S_1$ pulse signal $2\alpha$, Therefore, the transmission power and efficiency can be controlled by adjusting the voltage turn-on angles of $u_1$ and $u_2$.

As the LCC-S compensation circuit has a good filtering effect on the inverter output voltage, only the fundamental component is considered in the output voltage, and the effective value of the first-order component can be expressed as:

$$U_1 = \frac{2\sqrt{2}}{\pi} U_{\text{in}} \sin \alpha \tag{18}$$

Similarly, the RMS value of the first order component of the input voltage of the semi-controlled rectifier bridge is:

$$U_2 = \frac{2\sqrt{2}}{\pi} U_{\text{o}} \sin \beta \tag{19}$$

When the switching loss is ignored,

$$\frac{U_{\text{o}}^2}{R_{\text{L}}} = U_2 I_2 \tag{20}$$

The AC equivalent resistance of rectifier bridge can be obtained by combining Equations (19) and (20).

$$R_{\text{Leq}} = \frac{U_2}{I_2} = \frac{8}{\pi^2} R_{\text{L}} \sin^2 \beta \tag{21}$$

It can be seen from Equation (21) that when the DC load resistance is determined, only by changing the conduction angle of the semi-controlled rectifier bridge $\beta$, the AC equivalent resistance $R_{\text{Leq}}$ of the rectifier bridge can be changed accordingly. Therefore, in order to achieve the maximum efficiency of the tracking control system, the following equation needs to be satisfied:

$$\frac{8}{\pi^2} R_{\text{L}} \sin^2 \beta = R_{\text{best}} \tag{22}$$

Thus, the voltage conduction angle corresponding to the optimal transmission efficiency point can be derived:

$$\beta = \arcsin \sqrt{\frac{\pi^2 R_{\text{best}}}{8 R_{\text{L}}}} \tag{23}$$

In order to more comprehensively analyze the limiting conditions of the efficiency tracking control of the system, Equation (23) is substituted into Equation (15). According to the data in Table 1, using the parameter values in Table 1, the system efficiency, DC load resistance $R_{\text{L}}$, and conduction angle $\beta$ of the semi-controlled rectifier bridge can be obtained through simulation. The results are shown in Figure 7.

**Table 1.** The main parameters of the magnetic coupling resonant dynamic wireless power transmission system model.

| Parameter | Value |
| --- | --- |
| Coil internal resistance on receiving side and transmitting side $R_{1,2}$ | 0.13/0.13 Ω |
| Receiving side and transmitting side coil inductance $L_{1,2}$ | 108.47/108 µH |
| Compensation capacitance on receiving side and transmitting side $C_{1,2}$ | 114/93.8 nF |
| Frequency $f$ | 50 kHz |
| Compensation resonant coil inductance $L_{\text{f}}$ | 20 µH |
| Transmitting side parallel compensation capacitor $C_{\text{f}}$ | 507 nF |
| Compensation of internal resistance of resonant coil $R_{\text{f}}$ | 0.1 Ω |

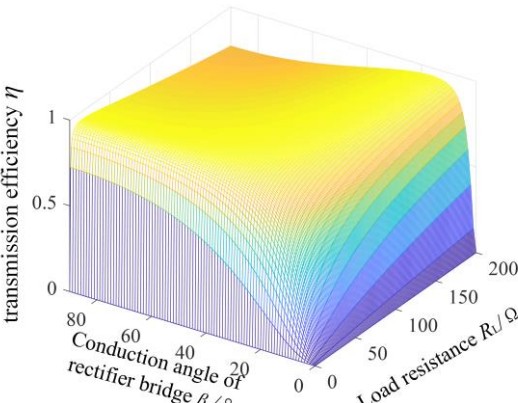

**Figure 7.** The relationship between $\eta$, $R_L$, and rectifier bridge conduction angle $\beta$.

By analyzing Figure 7, it can be concluded that within the scope of identification, each load resistance will correspond to a voltage conduction angle that can maximize the efficiency of the system. Therefore, when the load changes, by adjusting the voltage conduction angle corresponding to the semi-controlled rectifier bridge, the equivalent load $R_{Leq}$ is always equal to the optimal equivalent load $R_{best}$; the system efficiency can be maintained at the maximum. In addition, when the load value is small, the optimal efficiency cannot be achieved by adjusting the voltage conduction angle. It can be seen that the control of the voltage conduction angle of the rectifier bridge is limited. It can be seen from Equation (21) that when the load value is very small, even if the conduction angle is adjusted to the maximum angle, the equivalent resistance $R_{Leq}$ can only reach $8/\pi^2$ times the actual load value. In order to satisfy the optimal control of efficiency, the actual load $R_L$ of the system should satisfy the following equation:

$$R_L \geq \frac{\pi^2}{8} R_{best} \tag{24}$$

As with the principle of maximum efficiency tracking control, the expression of the voltage conduction angle at the transmitting side can be obtained from (14) and (18)

$$\alpha = \arcsin \sqrt{\frac{\pi^2 P_o \left[ M^2 R_f \omega + (L_f^2 \omega^2 + R_1 R_f)(R_2 + R_{Leq}) \right]^2}{8 R_{Leq} L_f^2 M^2 \omega^4 U_{in}^2}} \tag{25}$$

In the above equations, the output power $P_o$ is the known target power. When the mutual inductance $M$ or the load resistance $R_L$ changes, the transmitting side voltage conduction angle $\alpha$ can be controlled to keep the output efficiency at the specified value.

Based on the above analysis, this paper proposes a control strategy of bilateral adjustment to the transceiver side of a WPT system with an LCC-S compensation network. In the receiving side, the semi-controlled rectifier circuit is used to replace the traditional uncontrolled rectifier circuit. According to the change in the mutual inductance and load, the on and off of the switch devices in the rectifier bridge are adjusted to realize the efficiency optimization control of the system. In the transmitting side, the voltage conduction angle is changed by adjusting the internal phase shift angle added in the high-frequency inverter control circuit to achieve the purpose of regulating the output power.

Figure 8 shows the principle block diagram of bilateral control of the system. The control system mainly includes the control module at the transceiver side, current and voltage sampling, and other parts. The specific control flow is shown in Figure 9.

Before the system is running, it is initialized to make $\alpha = \alpha_0$ and $\beta = \beta_0$. $\alpha_0$ and $\beta_0$ can be between $0°$ and $90°$. When the system reaches a stable state, the DC output voltage $U_o$ and current $I_o$ are detected by the current and voltage sampling circuit at the receiving side to obtain the load resistance $R_L$. Secondly, the mutual inductance $M$ is substituted into

Equation (17) to obtain the optimal equivalent load $R_{best}$, and the receiving side controller updates the voltage conduction angle $\beta$ according to Equation (23).

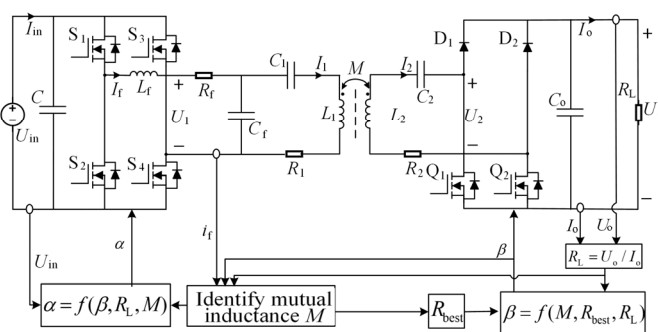

**Figure 8.** Control system block diagram.

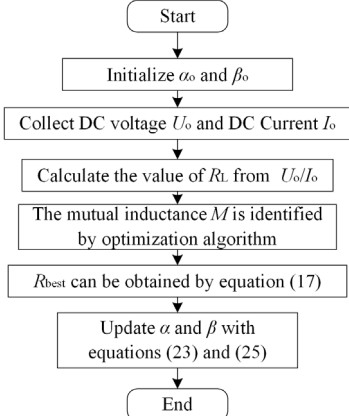

**Figure 9.** Control flow chart.

According to the analysis of transmission characteristics in Section 2.1, the mutual inductance parameter is one of the main factors affecting the output power and transmission efficiency of the system. Therefore, the adjustable range of mutual inductance should also be considered when regulating the transmission efficiency. The square coil with a side length of 30 cm is selected as the research object. Figure 10 shows the change in mutual inductance parameters of the receiving coil in the process of 0 cm to 25 cm lateral offset. The variation in the self-inductances of $L_1$ and $L_2$ is ignored in the optimization method because the variation of the self-inductances of $L_1$ and $L_2$ has a negligible effect on the system.

It can be seen from Figure 10 that with the increase in the transverse offset distance of the receiving coil, the mutual inductance between the transmitting coil and the receiving coil will drop rapidly. After that, this paper continues to study the influence of the lateral offset distance of the receiving coil on the output characteristics of the system. According to Equations (14) and (15), the changes in the system output power and transmission efficiency when the receiving coil is laterally offset are obtained, as shown in Figure 11.

According to Figure 11, when the transceiver coils are in the opposite position, the output power and transmission efficiency of the system are at the maximum point. With the gradual increase in the offset distance between the transceiver coils, the output power and transmission efficiency are in a downward trend, and the drop of output power is more obvious.

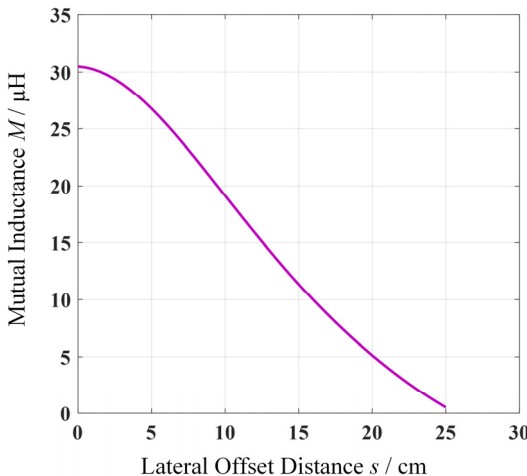

**Figure 10.** Change in mutual inductance during lateral shift.

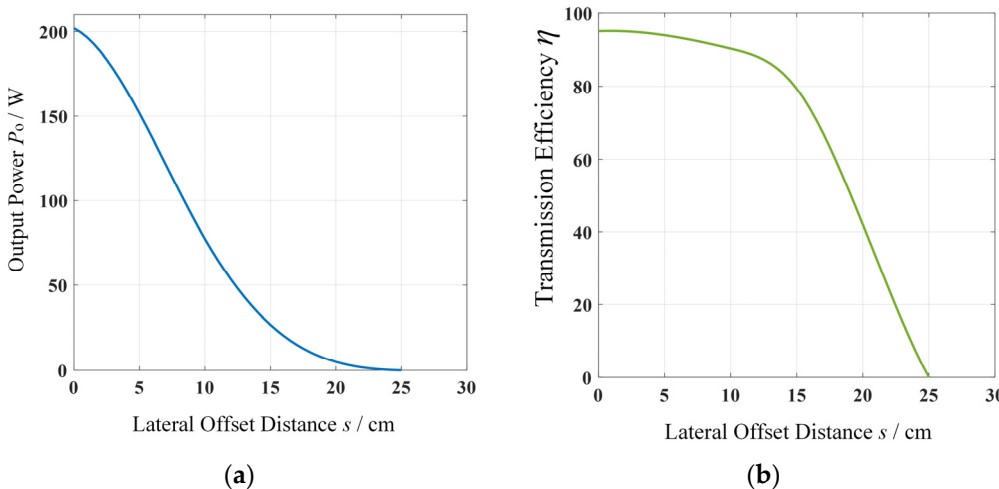

(**a**)                                        (**b**)

**Figure 11.** Changes in $P_o$ and $\eta$ under lateral offset. (**a**) Changes in output power $P_o$ under lateral offset and (**b**) changes in transmission efficiency $\eta$ under lateral offset.

Based on the above analysis of transmission characteristics, it is known that the adjustable range of mutual inductance needs to be considered when adjusting the efficacy under the change in mutual inductance parameters. Therefore, when the DC load $R_L$ is 30 Ω and the output power is 150 W, the relationship between the inverting voltage conduction angle $\alpha$, the rectifier voltage conduction angle $\beta$, and the mutual inductance parameter $M$ can be obtained by using the Equations (23) and (25), as shown in Figure 12.

As shown in Figure 12, there are several combinations of voltage conduction angles $\alpha$ and $\beta$ corresponding to each mutual inductance parameter when the stable output of the system power is satisfied. In order to determine the adjustable mutual inductance range corresponding to stable output power under optimal efficiency, the relationship between the rectifier voltage conduction angle $\beta$ and mutual inductance parameter $M$ when the $R_{Leq}$ of the AC equivalent load is equal to $R_{best}$ of the optimal AC equivalent load is obtained by using Equation (23). It can be seen that the intersection of the two surfaces is a curve, which represents the optimal voltage conduction angle $\alpha$ and $\beta$ when both the system's optimal efficiency and stable output power are satisfied at the given mutual inductance. When the mutual inductance is 12 μH, the conductance angle $\alpha$ of the inverted voltage has reached the maximum adjustment angle 90°. Combined with the mutual inductance measurement under the actual offset, the adjustable range of mutual inductance can be obtained at the output power of 150 W, which is 12 μH, 30 μH.

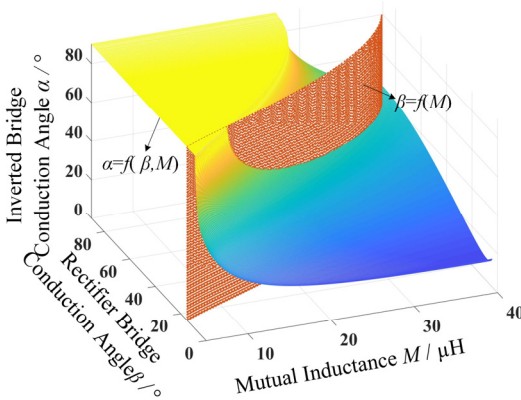

**Figure 12.** The relationship between $\alpha$, $\beta$, and *M*.

### 3.3. Simulation Analysis

In order to verify the feasibility of the proposed bilateral control strategy, a control model is built under the environment of Matlab/Simulink and simulated. When the system output power is set to 150 W, the power and efficiency changes in the semi-controlled rectifier, full-controlled rectifier, and uncontrolled rectifier circuits are compared under different mutual inductance values when the transmitter uses full-bridge inverted structure, as shown in Figure 13.

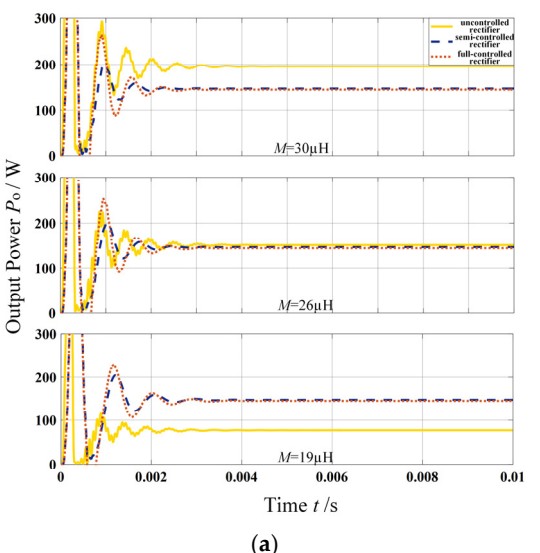

(**a**)

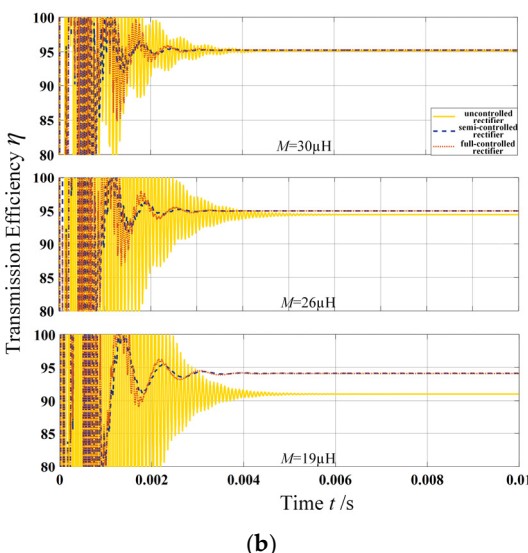

(**b**)

**Figure 13.** Changes in power and efficiency of different mutual inductances. (**a**) change in power under different mutual inductance and (**b**) change in efficiency under different mutual inductance.

In Figure 13, when the load resistance is 30 Ω and mutual inductances are 19 μH, 26 μH and 30 μH, the output power of semi- and full-controlled rectifiers is well controlled and can always be maintained at around 150 W. Compared with the control system, the output power of the system in uncontrolled rectifier mode decreases rapidly when the mutual inductance decreases, which does not meet the requirements of stable output power. In terms of efficiency control, the semi-controlled rectifier and full-controlled rectifier both show better control effect than uncontrolled rectifier mode. At the same time, since the functions of full-controlled rectifier and semi-controlled rectifier are similar, they work in the pure resistance state. However, the control drive circuit of the full-controlled rectifier is more complex, and the latter is cheaper and more efficient than the former. Therefore, it is more appropriate to select the topology of semi-controlled rectifier for secondary rectification.

Subsequently, the waveforms of the system output voltage $U_o$ with and without bilateral control are compared, as shown in Figure 14.

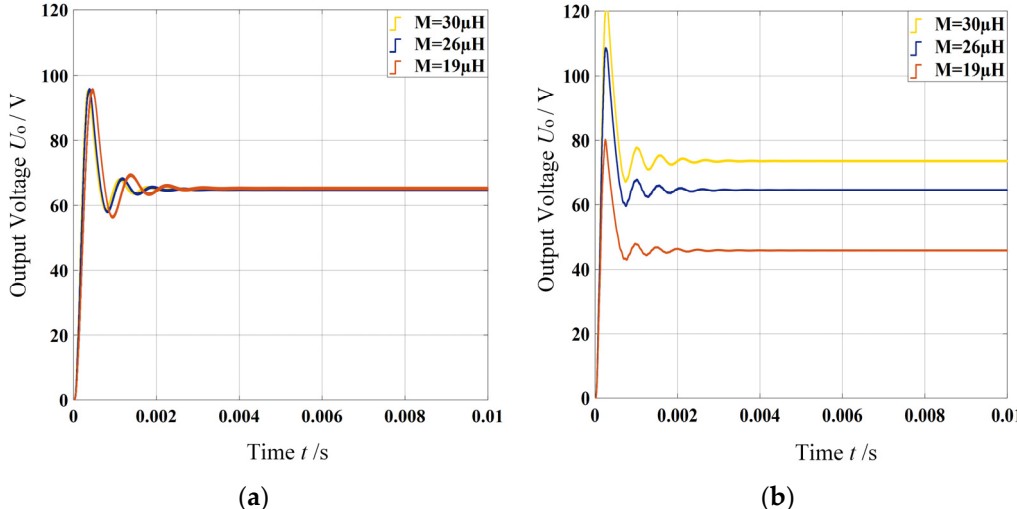

**Figure 14.** Change waveform of $U_o$ when control is applied and when control is not applied. (**a**) Variable waveform of $U_o$ under control and (**b**) waveforms of $U_o$ change when non-control is applied.

As shown in Figure 14, when non-control is applied to the system, the output voltage decreases with the decrease in mutual inductance. After applying bilateral control, the output voltage can be stabilized to about 65 V even when the mutual inductance value is gradually reduced, which enables the system to maintain a constant voltage output. Since the harmonic components can distort the current, which leads to a decrease in system efficiency, the FFT harmonic components of the transmit and receive side currents $i_f$, $i_1$, and $i_2$ are analyzed in the simulation environment, and the degree of distortion of the transmit and receive side currents in the operation of the system is recorded under the control and non-control, as shown in Table 2.

**Table 2.** Total harmonic distortion (THD) of $i_f$, $i_1$, and $i_2$ with and without control.

| Parameter | $M = 30\ \mu H$ | $M = 26\ \mu H$ | $M = 19\ \mu H$ |
| --- | --- | --- | --- |
| Exert control $i_f$ | 11.74% | 10.88% | 7.69% |
| Without control $i_f$ | 21.80% | 28.63% | 51.77% |
| Exert control $i_1$ | 0.88% | 1.08% | 0.80% |
| Without control $i_1$ | 1.58% | 1.23% | 0.78% |
| Exert control $i_2$ | 3.85% | 5.11% | 5.15% |
| Without control $i_2$ | 10.34% | 10.10% | 9.84% |

From the data recorded in Table 1, it is known that when the inverted output current $i_f$ is not controlled, the higher harmonic content of the current increases with the decrease in the mutual inductance parameter, which makes the distortion rate more serious; this change will produce more parasitic loss and reduce the transmission power of the system. When the control is applied, the distortion of the inverted output current $i_f$ during the mutual inductance change is significantly lower than that of the data when non-control is applied. At the same time, the distortion of emitter side current $i_1$ and receiver side current $i_2$ after applying control is also less than that when non-control is applied. It can be seen that adding bilateral control can reduce the degree of current distortion to a certain extent, so as to reduce the loss caused by harmonics and improve the energy efficiency of the system.

Finally, the variation in system output power and transmission efficiency with load resistance under different mutual inductance parameters is analyzed, as shown in Figure 15.

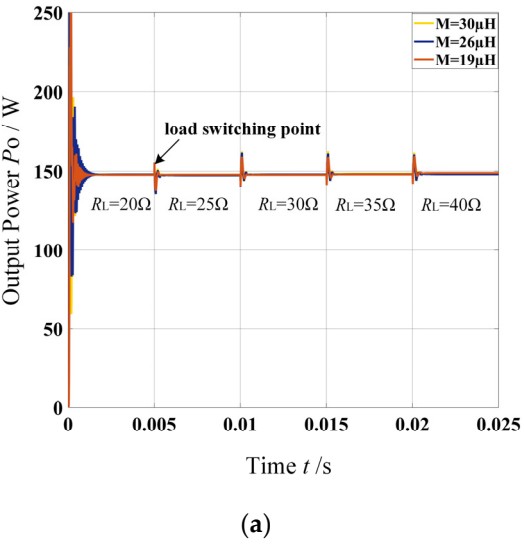

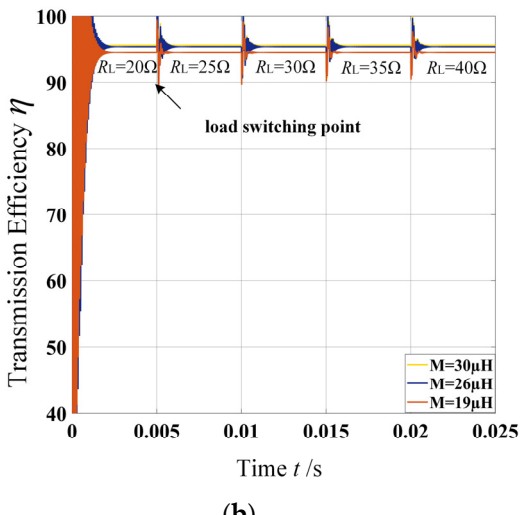

(**a**)　　　　　　　　　　　　　　　　　　　　　(**b**)

**Figure 15.** Changes in $P_o$ and $\eta$ under different $M$ and $R_L$. (**a**) Changes in output power $P_o$ under different $M$ and $R_L$ and (**b**) changes in efficiency $\eta$ under different $M$ and $R_L$.

From Figure 15a, it can be seen that the output power of the system can always be maintained at about 150 W with the increase in load resistance under the three states of mutual inductance: 30 µH, 26 µH, and 19 µH, respectively. Within the range of mutual inductance adjustable, the maximum and minimum power achievable are 151.85 W and 146.7 W, with fluctuations of 0.87% and 2.6%, respectively. In Figure 15b, when the mutual inductance is 30 µH and 26 µH, the transmission efficiency of the system is basically maintained at 95% with the increase in load resistance. When the mutual inductance is 19 µH, the efficiency decreases, but it also maintains at least 90%.

## 4. Experimental Verification

Based on the control framework designed in Section 3.2, an open-loop control system is set up. The parameters of each component of the circuit are shown in Table 1, and the specific experimental device is shown in Figure 16.

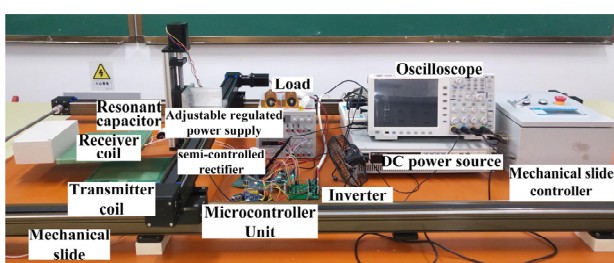

**Figure 16.** Experiment platform.

During the experiment, the DC input voltage of the control system is always kept at 50 V, the system output power is set to 150 W, the resonance frequency is 50 kHz, and the compensation networks are matched to the full resonance state.

To verify the control effect of system efficiency optimization and stable output power, the output states of the system are compared at different offsets while the distance between transmitting and receiving coils is 10 cm. Combining the mutual inductance adjustable range mentioned in Section 3.3 with the actual situation, the range of the offset distances of the transceiver coil is set here from 0 cm to 12 cm. Table 3 shows the simulation results and experimental results of the variation of system output power and transmission efficiency with offset distance under the bilateral control mode when the load resistance is 30 Ω, 35 Ω, and 40 Ω, respectively. Figure 17 shows the specific trend of system output power and

transmission efficiency with offset distance under two-sided control when load resistance is 30 Ω, 35 Ω, and 40 Ω, respectively.

**Table 3.** Simulation results and experimental results of transmission efficiency and received power under different load resistors.

| Number | 1 | 2 | 3 | 4 | 5 | 6 | 7 | 8 | 9 |
|---|---|---|---|---|---|---|---|---|---|
| Lateral offset distance $s$/cm | 0 | 6 | 12 | 0 | 6 | 12 | 0 | 6 | 12 |
| Load resistance $R_L$/Ω | 30 | 30 | 30 | 35 | 35 | 35 | 40 | 40 | 40 |
| Simulation results of output power $P_o$/W | 147.7 | 148.5 | 147.7 | 145.6 | 145.2 | 145.3 | 145.5 | 145.4 | 146.7 |
| Simulation results of transmission efficiency $\eta$ | 0.869 | 0.873 | 0.869 | 0.856 | 0.854 | 0.855 | 0.856 | 0.855 | 0.863 |
| Experimental results of output power $P_o$/W | 143.2 | 139.2 | 138.2 | 138.2 | 136.4 | 136.1 | 138.3 | 139.0 | 139.2 |
| Experimental results of transmission efficiency $\eta$ | 0.823 | 0.821 | 0.827 | 0.822 | 0.810 | 0.812 | 0.808 | 0.812 | 0.823 |
| Error between the simulation and experimental results | 0.030 | 0.062 | 0.064 | 0.051 | 0.061 | 0.063 | 0.052 | 0.044 | 0.051 |

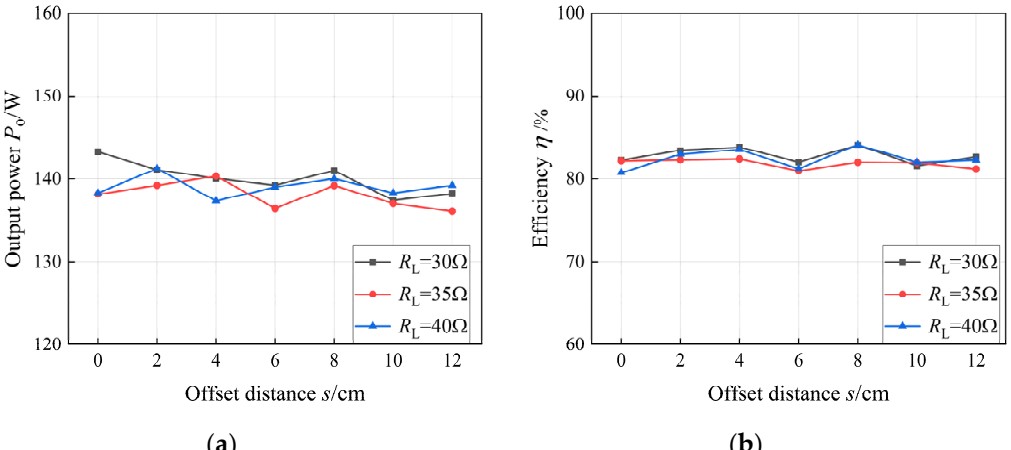

(**a**)  (**b**)

**Figure 17.** Changes in output power and system efficiency at different offsets. (**a**) Changes in output power at different offsets and (**b**) changes in system efficiency at different offsets.

The results show that there is little difference between the simulation results and the experimental results. Due to the energy loss in the real experiment, the simulation results are basically consistent with the experimental results. As can be seen from Figure 17, with the change in the offset distance, when the load resistance is 30 Ω, the maximum output power of the system is 143 W, the minimum value is 137 W, the average value is 139.9 W, and the fluctuation rate is about 2.21%. The maximum transmission efficiency is 84.1%, the minimum is 81.58%, the average is 82.84%, and the fluctuation rate is about 1.52%. When the load resistance is 35 Ω, the maximum output power of the system is 140 W, the minimum is 136 W, the average is 137.85 W, and the fluctuation range is from 1.34% to 1.56%. The maximum transmission efficiency is 82.4%, the minimum is 81%, the average is 81.86%, and the fluctuation range is between 0.66% and 1.05%. When the load resistance is 40 Ω, the maximum output power of the system is 141 W, the minimum is 137 W, the average value is 138.9 W, and the fluctuation range is from 1.37% to 1.49%. The maximum transmission efficiency is 84.13%, the minimum is 80.76%, the average is 82.41%, and the fluctuation rate is about 2%. Due to line loss and power meter measurement error, the control angle is biased, so there is a certain difference between the control effect and the simulation results. However, it can be seen that when mutual inductance and load parameters change, both the power and efficiency of the system can achieve stable output, and the system output voltage can maintain a constant output state under their respective loads.

## 5. Conclusions

Based on the LCC-S-type magnetically coupled resonant radio power transmission system, this paper conducts in-depth discussion and simulation modeling from three aspects: circuit theoretical characteristics and control strategy design, completing the experimental verification by setting up an experimental platform.

1. Based on the LCC-S type topology, the model is analyzed. On this basis, the effects of mutual inductance, equivalent load and compensation inductance on the transmission characteristics of dynamic wireless power transmission system are studied, which provides guidance for the selection of system parameters.

2. Considering the power and efficiency fluctuations in the operation of dynamic wireless power transmission system, and considering the energy transmission characteristics of the system, each voltage conduction angle is introduced into the inverter circuit and the semi-controlled rectifier circuit. In the adjustable range of mutual inductance, the efficiency optimization and power stable output control of the system are achieved by adjusting the degree-of-freedom of control on the transmitter side and receiver side. Both simulation and experimentation verify the validity and feasibility of the proposed bilateral control strategy.

**Author Contributions:** Writing—original draft preparation, M.X. and S.M.; supervision, Q.Y.; software, C.L.; writing—review and editing, P.Z.; validation, X.Z. All authors have read and agreed to the published version of the manuscript.

**Funding:** This research was funded by the National Natural Science Foundation of China and Tianjin Natural Science Foundation, grant number 52077153, 18JCQNJC70500 and 20JCYBJC00190.

**Conflicts of Interest:** The authors declare no conflict of interest.

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
