# Peer review of "Collaborative Optimization Method of Power and Efficiency for LCC-S Wireless Power Transmission System"

_electronics, doi:10.3390/electronics10243088_

Round 1

Reviewer 1 Report

The paper proposes a new topology of a Wireless Power Transmission (WPT) to optimize power transfer and its associated efficiency. The solution put forward is potentially interesting. Several basic issues have to be mentioned from the very beginning. The initial hypothesis and parameters of the problem are not completely stated. Firstly, the formulation of the problem seems incomplete, since the authors enter into details without clearly indicating the context and setting of the design. The reader quite abruptly learns, only from the context, that the subject is about powering wirelessly a vehicle (car, train, etc.), without knowing the specific mechanical parameters (mass, speed range, necessary active torque, type of suspension-classical/levitation, etc.) nor the type of electrical motor and its nominal parameters, etc. used for propulsion. Secondly, as mentioned before, the expected speed range of the vehicle is not specified. Having in view that the power is transmitted via the electromagnetic induction phenomenon, as Faraday’s law states, the induced emf has two components. The paper does not take into account the second one, namely the motional emf produced by the vehicle’s movement (at a certain speed). The induced emf is equated as if both components (the emitter and the receiver) were at rest. Thirdly, the emitting blocks (each comprising a high-frequency power source, a rectifier, an inverter, and a compensation circuit) are repetitively placed along the vehicle's running track. That structure would amount to hundreds of thousands of these identical building blocks to cover a reasonably long track for an electric-powered train transportation solution. Is it feasible economically? Are these resources available in terms of supply chain and production overall cost? Another concern arises. How to enable the appropriate emitting cell to switch on exactly at the right moment when the receiving cell passes above (at a certain speed-maybe not negligible). This technical problem is not a straightforward one, not being addressed within the paper, only stating that there is an “external sensing and control circuit”. To summarize, a lot of operation principle, technical and economic issues are still to be properly explained by the authors to justify the proposed approach. In more detail, the following additional remarks can be made.

  1. All the abbreviations must be defined for the reader’s convenience.
  2. The English text must be thoroughly revised for both style and grammar. There are numerous occurrences of subject-predicate mismatch throughout the text, typos, improper use of prepositions, capital letters, etc.
  3. Which “robot” do the authors mention on page 2 line 108? The vehicle is powered by a DC motor.
  4. Which is the role played by the battery in the schematic shown in Fig. 1?
  5. The load of the transport system is the DC motor. The equivalent circuit of such a motor cannot be reduced to a simple resistance RLeq, as shown in Fig. 2. There is a more complex representation comprising the induced back emf of the motor. The equivalent scheme must take into account the motor’s variable speed, accountable as variable equivalent circuit parameters, not constant ones.
  6. How is transmission efficiency defined in (15)?
  7. The flowchart shown in Fig. 9 appears twice, being present also in the previous Fig. 8.

Reviewer 2 Report

The paper proposes an interesting solution of obtaining stable output control of the wireless power system by controlling the conduction angles when the mutual inductance varies and considering different load resistances. The theoretical aspects are implemented in numerical modelling and then validated by experiment. In order to improve the quality of the paper I have the following remarks: 1) Please detail what means LCC-S (never explained in the paper); moreover, abbreviations are generally not recommended in the title of the paper. 2) Row 47: Please detail what means DD coils. 3) Row 95: you speak about “receiving and receiving side” – please correct. 4) Rows 126-127: something is not OK, maybe 127 must be inserted in front of 123, because it refers to figure 2. 5) Rows 144: instead of resistance it is better to speak about impedance (Zr). 6) There is a misunderstanding between elements from circuit from Figure 3 and relations (7) and (8) – please check. 7) Row 163: Please explain the assumptions from (10). 8) Row 171: please define the transmission efficiency η before using it. 9) Row 220: please check the phrase, something is not clear. 10) Rows 294 and 295 are identical! 11) There is no information about the optimization algorithm regarding the mutual inductance M identification, nor about the starting point in the optimization process α0 and β0 – please detail in the paper! 12) Please check “receiving and receiving”.

Reviewer 3 Report

This manuscript describes the collaborative optimization method of power and efficiency for LCC-s WPT systems. Although the idea is quite interesting, the manuscript needs major improvement in terms of writing organization and results. Please have my comments below: 

1) The writing structure of the abstract needs to be improved. In the abstract, please check the word selection and sentence structure. It is not clear what the authors mean by “aggravate the instability of the system” (page 1, line 14). Using long sentences and putting everything in it does not really help to explain the idea of the work (page 1, line 14-18). Overall, the main purpose and value proposition of the work are missing in the abstract. 

2) In the introduction, it is not clear what the authors try to explain on page 1, lines 40-45. Please put some special effort into sentence organization. Break it down into smaller segments. 

3) In page 2, lines 81-85, the authors stated that the previous work has been done based on a single performance index? What is this performance index? However, in the same sentence, the authors mentioned: “while considering the transmission efficiency and receiving power of WPT supply system.” The authors also did the same thing in this manuscript. So, what’s new here? 

4) For figure 4(a) and (b), what values of Lf, Rf and R1 have been considered? Do the values of these parameters affect the analysis? Please explain. 

5) On page 6, line 28, it is not clear the reason for adding an extra degree of freedom on the receiver side. Why the half- controlled rectifier is better than other methods to adjust the load resistance? For example, the authors can add the variable resistance in series with the load resistance and then adjust the optimal value. Please explain in detail. 

6) Experimental validation requires more result demonstration. How does the calculated power and efficiency for different load resistance match with that of simulation? Please add a table/ graph explaining the matching/difference. 

Round 2

Reviewer 1 Report

The paper has significantly gained in clarity and quality of the presentation. The authors have provided satisfactory responses to most of my remarks. There is a single further remark, namely to enhance the visibility of Figs. 1, 8, 9, and 12 (graph and drawings quality, font size, etc.), to name just the most critical ones. As a general remark, all the figures must be realized at an increased resolution. The graphs and drawings are fuzzy and foggy. Also, a final check of the English text is necessary.

Reviewer 3 Report

1. Check the spelling of "rectifier" in the manuscript.

2. All the picture quality of the updated manuscript is quite poor. Please check and update.

3. The reviewer is not still satisfied with table 3. Obviously, the readers should not look at Figure 17 for experimental and Table 3 for simulation results. Please add the following rows in Table 3:  
i) Experimental results of output power
ii) Experimental results of transmission efficiency
iii) % error between the simulation and experimental results.
